# An Assessment of the Effective Pollination Period and Its Main Limiting Factor in *Wurfbainia villosa* var. *villosa* (Lour.) Škorničk. & A. D. Poulsen (Zingiberaceae)

**DOI:** 10.3390/biology14091134

**Published:** 2025-08-27

**Authors:** Qianxia Li, Yanqian Wang, Ge Li, Shuang Li, Hongyou Zhao, Chunyong Yang, Zhibing Guan, Yating Zhu, Lin Xiao, Yanfang Wang, Lixia Zhang

**Affiliations:** 1Yunnan Key Laboratory of Sustainable Utilization of Southern Medicine, Yunnan Branch of Institute of Medicinal Plant Development, Chinese Academy of Medical Sciences and Peking Union Medical College, Jinghong 666199, China; 15925416204@163.com (Q.L.); yanqwang@implad.ac.cn (Y.W.); lige19800221@163.com (G.L.); k20071129@163.com (S.L.); hyzhao@implad.ac.cn (H.Z.); ace928@126.com (C.Y.); 13578155680@163.com (Z.G.); s2023009041@pumc.edu.cn (Y.Z.); 13308811908@163.com (L.X.); 2Yunnan Key Laboratory of Sustainable Utilization of Southern Medicine, Yunnan University of Traditional Chinese Medicine, Kunming 650500, China; 3Institute of Medicinal Plant Development, Chinese Academy of Medical Sciences and Peking Union Medical College, Beijing 100193, China

**Keywords:** *Wurfbainia villosa* var. *villosa*, stigma receptivity, style suitability, pollen tube growth rate, ovule longevity, initial fruit set, effective pollination period

## Abstract

*Wurfbainia villosa* var. *villosa* is cultivated for its fruits, which are mainly used as a traditional Chinese medicine and a cardamom-like spice. However, low fruit set is a problem, with the effective pollination period (EPP, defined as the window during which flowers can be fertilized) being a key factor. The duration of the EPP and its components was studied in two cultivars in the field over two years. The results showed that the EPP lasted two days, both from studies of its components (including stigma receptivity, style suitability, pollen tube growth rate, and ovule longevity) and from results of sequential pollination experiments. Style suitability was determined as the main limiting factor of EPP. These findings provide valuable insights for future research and could be applied to improve crop management.

## 1. Introduction

The procedure from pollination to successful fertilization is essential for fruit yield in plants and depends on the success of several sequential reproductive processes, including pollen transfer to receptive stigmas, pollen adhesion and germination on the stigma, pollen tube growth through the style, and, ultimately, ovule fertilization [1]. Issues with these processes can seriously impair fruit production, such as poor pollen viability, reduced stigma receptivity, or compromised ovule development. The concept of the effective pollination period (EPP) was first proposed by Williams (1965) [2,3], and is defined as the period during which pollination can successfully lead to fruit set. EPP is determined through Williams’ (1965) ‘indirect method’ [3], involving three physiological components assessed via fluorescence microscopy: stigma receptivity (SR), pollen tube growth rate (PTG, measured as the time required for pollen tubes to reach ovules), and ovule longevity (OL). EPP duration is calculated as OL minus PTG, provided that this value does not exceed the SR duration; otherwise, SR becomes the limiting factor determining EPP. Subsequent research on apricot [4] revealed that style suitability (SS) significantly influences pollen tube progression toward ovules and may consequently affect EPP duration. Williams (1970) [5] later introduced an alternative ‘direct method’, involving sequential hand-pollination of flowers at different post-anthesis intervals, where EPP duration corresponded to the pollination period between the extreme pollination dates achieving successful (earliest) or failed (latest pollination date) fruit sets. The EPP can help identify the time when the pollination is effective and help provide a good framework for the detection of factors limiting fruit set. Since its introduction, the EPP has been extensively studied across various fruit species, including apple [6,7], pear [7], apricot [8], kiwifruit [9], sweet cherry [10], peach [11], and olive [1] under diverse environmental conditions. Notably, the EPP has a greater impact than flower number on the fruit crop, and plays a clear part in explaining erratic or low production [5].

*Wurfbainia villosa* var. *villosa* (Lour.) Skornick. and A. D. Poulsen (homotypic synonym: *Amomum villosum* Lour.) [12] is a perennial evergreen herb in the family Zingiberaceae and an obligate shade plant native to humid subtropical regions, including South China and Southeast Asia (Thailand, Vietnam, Laos, and Cambodia) [13,14,15]. Its dried ripe fruits, known as *sharen* in Chinese, are a primary source of the traditional Chinese medicine Amomi Fructus (AF), as documented in the Chinese Pharmacopoeia (2020) [16]. With a history of over 1300 years in traditional medicine, AF has been widely used to treat gastrointestinal and pregnancy-related disorders [17,18,19,20]. In China, Guangdong and Yunnan Provinces are the *daodi* (authentic production regions) and primary cultivation areas for *W. villosa* var. *villosa*, accounting for over 90% of the national output. However, low fruit set severely limits AF production, with natural pollination yielding only 1.23–14.24% initial fruit set (IFS) in Guangdong [21] and less than 10% in Yunnan (based on our field surveys).

The floral organ of *W. villosa* var. *villosa* has a special structure. The slender style is sandwiched between the two pollen sacs of the anther, providing upward growth support for the style. The funnel-like stigma has an upward opening and protrudes beyond the apex of the anther, which makes the stigma higher than the pollen and difficult to self-pollinate. The stigma and anther are half-enclosed in the lip, and the cleft of the pollen sacs is attached to the lip, so only very few insects can effectively pollinate [21]. Previous studies attributed the low IFS to a scarcity of effective pollinators and the short viability period of pollen and stigmas [20]. Although artificial pollination can improve IFS, critical aspects of the reproductive biology of *W. villosa* var. *villosa*—including stigma receptivity, style suitability, pollen tube growth dynamics, ovule longevity, pollen–pistil interactions, and the EP—remain poorly characterized. This knowledge gap impedes the optimization of field practices. Consequently, artificial pollination in current production practices is typically concentrated on the morning of the first anthesis day (based on our field surveys), resulting in high labor costs. Therefore, the specific objectives of this study were: (1) to determine the EPP by assessing stigma receptivity, style suitability, pollen tube growth rate, and ovule longevity using fluorescence emission microscopy and evaluating the initial fruit set (IFS) of flowers pollinated by hand at different days after anthesis in two cultivars of *W. villosa* var. *villosa* under natural field conditions in Xishuangbanna Dai Autonomous Prefecture, Yunnan Province, China; and (2) to identify the primary EPP component limiting the duration of EPP in *W. villosa* var. *villosa*. The determination of the EPP and its main limiting factor in *W. villosa* var. *villosa* will enable a better understanding of the role played by each component of pollen–pistil interaction in the productivity losses often occurring in IFS, and contribute to optimizing the artificial pollination window, thereby maximizing labor efficiency.

## 2. Materials and Methods

### 2.1. Experimental Site and Plant Material

The EPP of *W. villosa* var. *villosa* was determined by two cultivars, namely, ‘Yunsha 1 Hao’ and ‘Yunsha 2 Hao’, in 2022 and 2023 at the Southern Medicinal Plant Garden of Yunnan Branch Institute of Medicinal Plant Development, Xishuangbanna Dai Autonomous Prefecture, China (22°0′21.04″ N, 100°47′17.71″ E, altitude 556 m).

The two cultivars were planted in 2017 using ramet seedlings and were maintained under field conditions with standard watering and fertilization practices [22]. The plants were in their main harvest years (typically spanning 10–15 years after planting). EPP was determined by investigating stigma receptivity, style suitability, pollen tube growth rate and ovule longevity using a microscopic fluorescence approach, and by the initial fruit set (IFS) by hand-pollinating flowers at different days after anthesis.

### 2.2. Determination of Effective Pollination Period Using Stigma Receptivity, Style Suitability, Pollen Tube Growth Rate, and Ovule Longevity

#### 2.2.1. Methods for Assessing Stigma Receptivity and Style Suitability

Stigma receptivity and style suitability were determined on 30 flowers per sampling date. They were enclosed in gauze bags before anthesis to prevent unwanted pollination. Both cultivars were cross-pollinated using a paintbrush loaded with fresh pollen grains of the same cultivar between 09:00 and 11:00 from 0 to 2 DAA, as shown in Figure 1. Before pollination, pollen viability was confirmed to exceed 90% in both cultivars by the TTC method [21]. Pollinated flowers were sampled 24 h after pollination (HAP) in 2022 and 48 h in 2023. The entire pistil (including the stigma, style, and ovary) was fixed in FAA solution (38% formaldehyde:acetic acid:70% ethanol = 1:1:8, by volume) and stored in the dark at room temperature for at least 48 h before further processing. Before microscopic observation, pistils were taken out from the FAA, rinsed for 2–3 min under tap water, and sequentially treated as follows: hyalinized with 1.6% NaClO for 3–4 h, softened with 12 M NaOH for 3–4 h, stained with 0.2% aniline blue in 0.1 M KH_2_PO_4_ buffer at least 36 h. After staining, the stigma and style were cut from the base of the style and placed on a microscope slide, while the ovary was cut longitudinally from the middle and placed on a separate microscope slide. A drop of staining solution was added onto the sample, and it was covered with a coverslip. The slide was pressed to disperse the tissue, exposing embedded pollen grains and pollen tubes for observation. Samples were examined under an Olympus IX73 microscope (Olympus Corporation, Tokyo, Japan) equipped with a wide band UV excitation (exciter filter BP340-390, dichroic beamsplitter DM410, barrier filter BA420-IF) and a 100-watt ultra-high pressure mercury lamp for fluorescence observation. The stigma receptivity was evaluated based on pollen adhesion (presence of pollen grains) and pollen germination (presence of germinated pollen tubes), which were rated on a binary absence/presence scale (0 = absence, 1 = presence), as previously described in apple [23]. Style suitability was also determined on a binary scale, based on whether the pollen tubes reached the ovules (0 = no, 1 = yes) [24]. Data were analyzed with a binary logistic analysis under a generalized linear model (GLM) (with the percentages of flowers with pollen adhesion, pollen germination, pollen tubes reaching the lower style and entering the ovules as the dependent variable, and different pollination days after anthesis of each cultivar as the factor).

All chemical reagents used in the experiments, including TTC, formaldehyde, acetic acid, ethanol, NaClO, NaOH, and KH_2_PO_4_, were sourced from Tianjin Damao Chemical Reagent Co., Ltd. (Tianjin, China).

#### 2.2.2. Methods for Assessing Pollen Tube Growth Rate

The pollen tube growth rates of both cultivars were measured by cross-pollinating flowers with fresh pollen grains from the same cultivar at 0 DAA. Before anthesis, the flowers were isolated using gauze bags to prevent unwanted pollination and then hand-pollinated at 0 DAA. At 8, 24, and 48 HAP, 30 flowers were sampled, respectively. The entire pistil (including the stigma, style, and ovary) was fixed in FAA solution (38% formaldehyde:acetic acid:70% ethanol = 1:1:8, by volume) and stored in the dark at room temperature for at least 48 h before further processing. Before microscopic observation, samples were rinsed with tap water, hyalinized with NaClO, softened with NaOH, stained with aniline blue, and pistil sections were made, following the previously described procedure in Section 2.2.1 (Methods for Assessing Stigma Receptivity and Style Suitability). The slide was pressed to disperse the tissue, exposing embedded pollen tubes for observation. Pollen tube growth was assessed under a microscope, with the growth rate determined based on whether the tubes had successfully reached the ovules (0 = no, 1 = yes), as previously described in apple [23]. Data were analyzed with a binary logistic analysis under GLM (with the percentages of flowers with pollen tubes entering the ovules as the dependent variable, and different sampling times after pollination of each cultivar as the factor).

#### 2.2.3. Methods for Assessing Ovule Longevity

Ovule longevity of both cultivars was evaluated using unpollinated flowers isolated with gauze bags. A total of 30 flowers were sampled at each time point from 0 to 9 DAA. The ovaries were immediately fixed in FAA solution (38% formaldehyde:acetic acid:70% ethanol = 1:1:8, by volume) and stored in the dark at room temperature for at least 48 h before further processing. Before microscopic observation, samples were rinsed, hyalinized, softened, stained, and ovary sections were made, following the method previously described in Section 2.2.1 (Methods for Assessing Stigma Receptivity and Style Suitability). The slide was pressed to disperse ovules in the ovary into a single plane for observation. Fluorescent ovules are considered non-viable, as previously reported in olive [1]. Here, multi-ovulate flowers were evaluated using a binary scale (0 = non-fluorescent, 1 = fluorescent). Data were analyzed with a binary logistic analysis under GLM (with the percentages of flowers with fluorescent ovules as the dependent variable, and different days after anthesis of each cultivar as the factor).

### 2.3. Determination of Effective Pollination Period Assessed Using Initial Fruit Set

Initial fruit set (IFS) in response to sequential pollinations at 0,1,2 DAA was employed in the assessment of the EPP. In both cultivars, experimental flowers were isolated using gauze bags before anthesis, and 30 bagged flowers of consistent age were cross-pollinated at each date with fresh pollen grains of the same cultivar. IFS was recorded 10 days after pollination using a binary scale (0 = absence, 1 = presence). Data were analyzed with a binary logistic analysis under GLM (with IFS as the dependent variable, and different pollinated days after anthesis of each cultivar as the factor).

### 2.4. Statistical Analysis

All statistical analyses were conducted using IBM SPSS Statistics 22 (IBM, Armonk, NY, USA).

## 3. Results

### 3.1. Effective Pollination Period Assessed by Stigma Receptivity, Style Suitability, Pollen Tube Growth Rate, and Ovule Longevity

#### 3.1.1. Stigma Receptivity and Style Suitability

The stigma of *W. villosa* var. *villosa* has a funnel-like structure. We observed that the inner surface of the funnel serves as the receptive region, facilitating pollen adhesion and germination. Upon germination, the pollen forms a pollen tube, which grows through the stylar canal, enters the ovary, and ultimately reaches the ovule (Figure 2). In this study, pollen adhesion (determined by the presence of pollen grains) and pollen germination (the presence of germinated pollen tubes) were used to evaluate the stigma receptivity, while the pollen tube growth was assessed to determine the style suitability.

The results indicated relatively stable pollen adhesion from the time of pollination (Table 1). In ‘Yunsha 1 Hao’, the percentage of flowers with attached pollen showed no significant decline from 100% (at 0 or 1 days after anthesis, DAA) to 89.5% (at 2 DAA) in 2022 (Wald χ2 = 2.235, df = 2, *p* = 0.327), and to 90.0% (2 DAA) in 2023 (Wald χ2 = 3.333, df = 2, *p* = 0.189). More pronounced decreases were observed in ‘Yunsha 2 Hao’, where the percentage of flowers with pollen attached did not differ significantly between 82.6 and 100% (at 0–1 and 2 DAA, respectively) in 2022 (Wald χ2 = 4.842, df = 2, *p* = 0.089), but dropped significantly from 100% (at 0 or 1 DAA) to 40.3% (at 2 DAA) in 2023 (Wald χ2 = 39.231, df = 2, *p* < 0.001). Pollen germination decreased more markedly as pollination was delayed. In ‘Yunsha 1 Hao’, the percentage of flowers with germinated pollen dropped significantly as the date of manual pollination was delayed, from 100% (at 0 or 1 DAA) to 63.2% (at 2 DAA) in 2022 (Wald χ2 = 11.083, df = 2, *p* = 0.004), and to 73.3% (at 2 DAA) in 2023 (Wald χ2 = 10.909, df = 2, *p* = 0.004). A similar trend was observed in ‘Yunsha 2 Hao’, where the percentage of flowers with germinated pollen decreased significantly from 100% (at 0 DAA) to 78.3% (at 1 DAA) in 2022 (Wald χ2 = 7.514, df = 2, *p* = 0.023), and significantly from 100% (at 0 or 1 DAA) to 40.0% (at 2 DAA) in 2023 (Wald χ2 = 45.000, df = 2, *p* < 0.001).

Pollen tube growth also showed slower progression with later pollination (Table 1). Both cultivars exhibited optimal pollen tube growth when pollinated at 0 DAA. In ‘Yunsha 1 Hao’, the percentage of flowers with pollen tubes entering the ovules at 0 DAA was 95.2% (in 2022) and 93.3% (in 2023), but significantly decreased to 61.5% (Wald χ2 = 461.600, df = 2, *p* < 0.001) and 36.7% (Wald χ2 = 437.368, df = 2, *p* < 0.001) at 1 DAA, respectively. By 2 DAA, no pollen tubes reached the ovules or the lower part of the style when sampled at 24 h post-pollination (by which time the stigma and style had already begun to wilt) in 2022, while, in 2023, only 60.0% of pollen tubes reached the lower part of the style by the 48 h sampling point, at which point the stigma and style were completely wilted. ‘Yunsha 2 Hao’ exhibited similar trends, with 87.1% (2022) and 96.7% (2023) flowers having pollen tubes entering the ovules when pollinated at 0 DAA, figures which significantly declined to 50.0% (Wald χ2 = 235.250, df = 2, *p* < 0.001) and 70.0% (Wald χ2 = 257.506, df = 2, *p* < 0.001) at 1 DAA, and 0% and 6.7% at 2 DAA, respectively. At delayed pollination (2 DAA), 0% (2022) and 30.0% (2023) of pollen tubes were restricted to the lower part of the style at their respective sampling times.

Based on the above analysis, the stigmas remained capable of supporting pollen adhesion and germination until 2 DAA. Thus, stigma receptivity persisted for more than three days (0, 1, and 2 DAA). By contrast, the style had almost completely lost its ability to facilitate pollen tube entry into ovules by 2 DAA, indicating that style suitability was limited to two days (0 and 1 DAA), this latter component being markedly shorter than stigma receptivity.

#### 3.1.2. Pollen Tube Growth Rate

The earliest observation of the pollen tubes reaching the ovules was 8 h after pollination in both cultivars (Table 2). However, the percentage of flowers with pollen tubes reaching the ovules was low, with values of 8.3% (2022) and 3.2% (2023) in ‘Yunsha 1 Hao’, and 12.0% (2022) and 0% (2023) in ‘Yunsha 2 Hao’. The higher percentages were recorded 24 h after pollination with values of 88.9% (2022) (Wald χ2 = 136.383, df = 2, *p* < 0.001) and 100% (2023) (Wald χ2 = 930.000, df = 2, *p* < 0.001) in ‘Yunsha 1 Hao’, 75.9% (2022) (Wald χ2 =119.551, df = 2, *p* < 0.001) and 64.5% (2023) (Wald χ2 = 280.364, df = 2, *p* < 0.001) in ‘Yunsha 2 Hao’, and the percentages continued to rise in both cultivars at 48 h after pollination, although the differences were significantly from values at 24 h in only ‘Yunsha 2 Hao’, indicating that most pollen tubes reach the ovules within one day. Additionally, we observed that, although numerous pollen tubes grew down the style and entered the ovary, only one or two pollen tubes successfully reached each ovule (Figure 2).

#### 3.1.3. Ovule Longevity

Ovule longevity was defined as the period before fluorescence emission began; i.e., the period during which all ovaries contained 100% viable ovules. The bright fluorescence observed under microscopy on the vascular bundles of ovules served as an indicator of ovule senescence. However, this fluorescence was transient and eventually disappeared, making it difficult to distinguish degenerated ovules from viable ones (Figure 3). Therefore, in this experiment, we quantified ovule longevity by recording the percentage of flowers exhibiting bright fluorescence on ovule vascular bundles at each sampling day after anthesis.

The results (Table 3) showed that bright fluorescence first appeared on ovule vascular bundles at 3 DAA in ‘Yunsha 1 Hao’, persisted for four to five days, and then gradually faded. The highest percentages of ‘Yunsha 1 Hao’ flowers with fluorescent ovules occurred at 3 DAA in 2022 (75.0%) (Wald χ2 = 83.856, df = 2, *p* < 0.001) and 5 DAA in 2023 (96.4%) (Wald χ2 = 457.582, df = 2, *p* < 0.001). In ‘Yunsha 2 Hao’, fluorescence was first detected at 3 DAA in 2022 and 4 DAA in 2023, with highest percentages at 4 DAA in both years (50.0% in 2022 (Wald χ2 = 23.052, df = 2, *p* = 0.002) and 92.9% in 2023 (Wald χ2 = 381.320, df = 2, *p* < 0.001)). Based on these criteria, ovule longevity in both cultivars lasted at least 3 days (0–2 DAA).

#### 3.1.4. Estimated Effective Pollination Period

Building on Williams’ (1965) [3] original EPP model, we introduced style suitability (SS) as an additional constraint parameter. Our results demonstrated that stigma receptivity (SR) persisted for more than three days (0, 1, and 2 DAA), while SS was limited to two days (0 and 1 DAA), and the ovule longevity minus pollen tube growth rate (OL−PTG) duration exceeded two days in both ‘Yunsha 1 Hao’ and ‘Yunsha 2 Hao’. Critically, SS emerged as the primary EPP-limiting factor, being shorter than the durations of either SR or OL-PTG. Consequently, the EPP for both cultivars was constrained to two days (0 and 1 DAA) (Table 4).

### 3.2. Effective Pollination Period Assessed by Initial Fruit Set

In ‘Yunsha 1 Hao’ (Figure 4), the highest initial fruit set (IFS) rates were observed with values of 60.0–62.1% when pollinated at 0–1 DAA in 2022 (Wald χ2 = 12.881, df = 2, *p* = 0.002), and 83.3% at 0 DAA in 2023 (Wald χ2 = 7.131, df = 2, *p* = 0.028), respectively. Similarly, ‘Yunsha 2 Hao’ exhibited the highest IFS rates with values of 58.6–64.3% at 0–1 DAA in 2022 (Wald χ2 = 12.904, df = 2, *p* = 0.002), and 86.2% at 0 DAA in 2023 (Wald χ2 = 23.722, df = 2, *p* < 0.001). In both cultivars, the IFS rates at 2 DAA had significantly declined to 3.4–3.6% in 2022 and 0% in 2023, indicating that the EPP lasts only two days (0 and 1 DAA).

## 4. Discussion

In this study, the EPP of *W. villosa* var. *villosa* was determined in two cultivars for the first time by a component-based method assessing stigma receptivity, style suitability, pollen tube growth rate, and ovule longevity using fluorescence microscopy, and by evaluating initial fruit set (IFS) through sequential pollination. Our results show that stigma receptivity lasted more than three days, style suitability was maintained for two days, most pollen tubes reached the ovules within one day, and ovule longevity persisted for at least three days. The IFS was highest when pollinated at 0–1 days after anthesis (DAA) but dropped sharply to near 0% by 2 DAA. Both methods yielded an EPP of two days, with style suitability identified as the primary limiting factor. Accurately determining the EPP and its main limiting factor in *W. villosa* var. *villosa* provides a theoretical foundation for optimizing artificial pollination in current production practices. This extends the practical pollination window from just the morning of the first anthesis day (0 DAA) to the entire first day and until at least 11:00 on the following day (1 DAA), thereby reducing labor intensity and overall costs.

When assessing stigma receptivity using fluorescence microscopy, this study revealed for the first time that the receptive surface in *W. villosa* var. *villosa* is located on the inner surface of the funnel-shaped stigma (Figure 2). This finding contrasts with observations in many other plants, such as pear [25], sweet cherry [26], almond [27], apple [28,29], and olive [1], where the receptive surface is found on the outer side of the stigma. This unique structure made it more challenging to observe pollen adhesion and pollen germination on the stigma, as pollen grains clustered together inside the funnel-shaped cavity and became difficult to accurately count when the stigma was crushed (Figure 2b). Therefore, a binary absence/presence scale (0 = absence, 1 = presence) was set here to assess pollen adhesion and pollen germination.

Another concern about stigma receptivity in this study relates to the slightly modified indicators used to evaluate stigma receptivity by fluorescence microscopy, in which stigma receptivity was assessed based on pollen adhesion and germination, as in studies in olive [1,30]. The results showed that the stigma remained capable of supporting pollen adhesion and germination until 2 DAA, indicating that stigma receptivity persisted for more than three days (0, 1, and 2 DAA), longer than the two-day period reported previously using the benzidine-hydrogen peroxide method (characterized by blue-stained stigmas with abundant bubbles) [21]. Studies in pear [25,31] and almond [27] considered pollen germination and initial pollen tube growth to be more reliable indicators, as degenerate stigmas may still support pollen adhesion and germination [25]. However, accurately assessing initial pollen tube growth remains methodologically challenging in this study. Thus, the estimates of stigma receptivity duration determined here may be exaggerated.

To overcome the absence of standardized criteria for detecting initial pollen tube growth, we introduce ‘style suitability’ as a compensatory parameter. This metric quantifies the style’s ability to support pollen tube growth, the senescence of which might prevent the growth of pollen tubes on their way to the ovules, even though viable ovules are still present in the ovary [1,4]. Therefore, we determined style suitability based on the capacity of the style to support pollen tube growth to the ovules. Our results demonstrated that the style lost nearly all of this ability by 2 DAA (Table 1), indicating that style suitability was two days (represented by 0 and 1 DAA).

The pollen tube growth rate of most plants is typically evaluated during the early flowering stage when stigma receptivity is high [1,27]. In this study, the pollen tube growth rate in *W. villosa* var. *villosa* was observed for the first time using fluorescence microscopy. Pollen tube growth rate was assessed by manually pollinating flowers with fresh pollen grains from other flowers from the same cultivar at 0 DAA, at which point the stigma exhibited the greatest receptivity.

As one of the components of the effective pollination period (EPP), accelerated pollen tube growth rate can enhance EPP when the ovule viability period, minus the time required for pollen tubes to reach the ovules, exceeds the stigma receptivity and style suitability periods. Previous studies have reported that pollen tube growth rates vary significantly depending on species, cultivar, flower nutritional status, and environmental conditions [30,32]. Our results showed that, while the two cultivars exhibited different pollen tube growth rates, most pollen tubes in both cultivars reached the ovules within 24 h (one day) after pollination. This similarity may be attributed to the relatively long sampling intervals in this experiment (8 h, 24 h, and 48 h post-pollination). To determine a more precise time for pollen tube growth rate, future studies should incorporate shorter sampling intervals.

Fluorescence microscopy has been widely used to evaluate ovule longevity in various plant species, including plum [33], olive [1,34], sweet cherry [26], and apple [23]. In these studies, callose deposition on the ovule surface exhibited bright fluorescence when stained with aniline blue, serving as a reliable indicator of ovule senescence. In this experiment, we observed, for the first time, a similar bright fluorescence in *W. villosa* var. *villosa* using fluorescence microscopy. Unlike in plum [33], olive [1], and sweet cherry [26], where fluorescence appeared across the entire surface of senescent ovules, the fluorescence in *W. villosa* var. *villosa* was localized specifically along the vascular bundle of the ovule, extending from the chalaza to the placenta. Moreover, this fluorescence gradually diminished as the ovule aged, making it difficult to distinguish senescent ovules from viable ones at later stages (Figure 3). A comparable phenomenon was reported in olive, where ovules turned blackish and lost fluorescence in advanced stages of senescence [34].

To assess ovule longevity in *W. villosa* var. *villosa*, we adapted the method described for olive [34]. Flowers were sampled at one-day intervals from 0 to 9 DAA. Ovule longevity was determined as the last day before bright fluorescence appeared on the ovule’s vascular bundle; i.e., the period during which all ovaries contained 100% viable ovules. Based on this criterion, ovule longevity in both cultivars of *W. villosa* var. *villosa* lasted at least three days. However, this estimate may be shorter than the actual functional viability, as studies in sweet cherry [35] suggest that the ovule fluorescence might be more closely related to the aging of the ovule than the actual viability. Similarly, research on olive indicates that fluorescence might well be preceded by the actual loss of ovule functionality for some days [1].

In this study, both the component-based assessment and the IFS results indicated an EPP of two days, which contrasts with earlier studies on olive [1,24,30], where the EPP assessed through IFS was considered more accurate. The component-derived method based on Williams’ original EPP model may overestimate EPP due to wrong assessments of the duration of particular components, although it is useful to identify limiting factors [1]. As reported in olive, EPP was prolonged with style suitability (SS) omitted and stigma receptivity (SR) evaluated solely based on pollen adhesion and germination [1]. Here, we refined Williams’ (1965) [3] EPP model when there is a limitation in SS, calculating EPP as the minimum duration among SR, SS, and ovule longevity (OL) minus pollen tube growth rate (PTG) (EPP = min(SR, SS, OL−PTG). Crucially, SS was the primary limiting factor in our study, and its duration matched the IFS-derived EPP, confirming that our component-based assessment was reliable. However, the values of SR, SS, PTG, OL, and the IFS differed between 2022 and 2023 in both cultivars of *W. villosa* var. *villosa*. These differences are likely influenced by genetic, physiological, and environmental factors during flowering [2], which should be the focus of future studies.

## 5. Conclusions

In this study, for the first time, the EPP of *W. villosa* var. *villosa* was determined in two cultivars by measuring stigma receptivity, style suitability, pollen tube growth rate, ovule longevity using fluorescence microscopy, and by initial fruit set (IFS) of hand-pollinated flowers on different days after anthesis. Both the component-based assessment and the IFS results indicated an EPP of two days, and the style suitability was determined as the main limiting factor of EPP in *W. villosa* var. *villosa*. These findings are a valuable addition to future research and have applications to improve crop management. This extends the practical pollination window from just the morning of the first anthesis day (0 DAA) to the entire first day and until at least 11:00 on the following day (1 DAA), thereby reducing labor intensity and overall costs.

Our future research will study how environmental factors, particularly temperature and humidity, affect the EPP and its components in *W. villosa* var. *villosa*. This will provide practical guidance for field management and increase fruit set.

## Figures and Tables

**Figure 1 biology-14-01134-f001:**
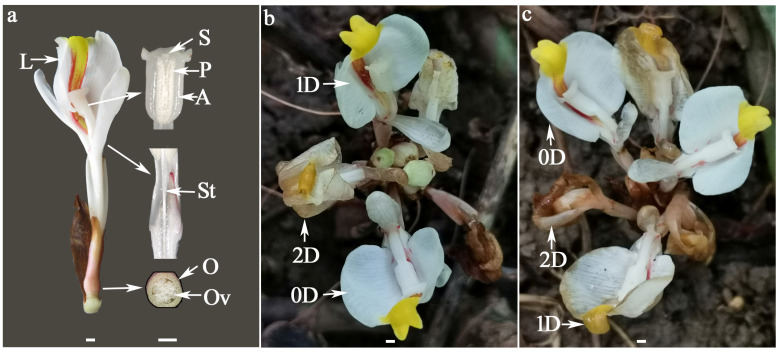
The floral structure of *W. villosa* var. *villosa* (**a**) and its morphological changes from 0 to 2 DAA on sunny (**b**) or rainy (**c**) days. A = anther, D or DAA = days after anthesis, L = lip, O = ovary, Ov = ovule, P = pollen, S = stigma, St = style; Scale bar = 2 mm.

**Figure 2 biology-14-01134-f002:**
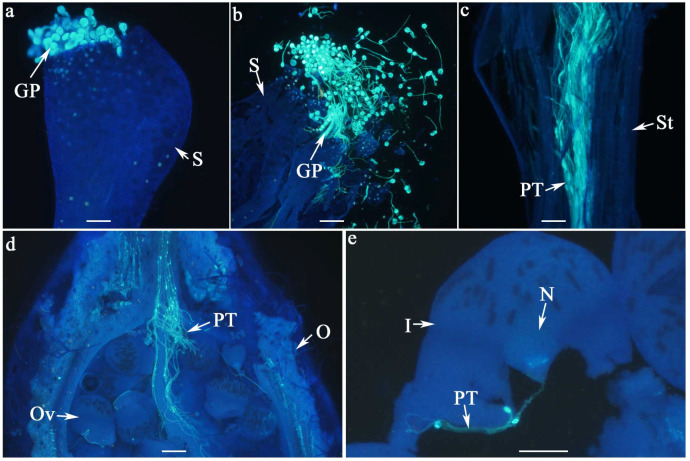
Pollen–pistil interaction in *W. villosa* var. *villosa*, as illustrated by fluorescence emission microscopy. A general overview of pollen adhesion and germination on the inner surface of the funnel-shaped stigma following pollination (**a**), pollen grains within the stigma are dispersed upon crushing the stigma (**b**), the pollen tube elongates along the stylar canal (**c**), penetrates the ovary (**d**), and ultimately reaches the ovule (**e**); GP = germinated pollen, I = integument, N = nucellus, O = ovary, Ov = ovule, PT = pollen tubes, S = stigma, St = style; Scale bar = 100 µm.

**Figure 3 biology-14-01134-f003:**
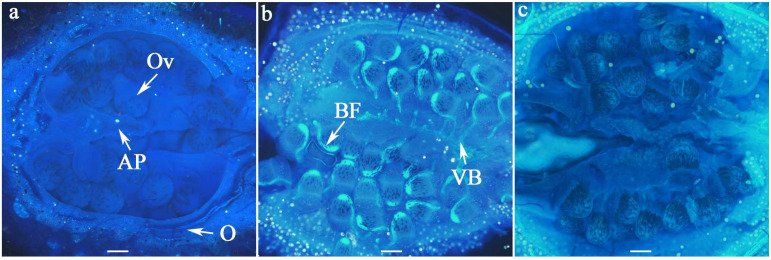
Fluorescence indicating non-viable ovules in longitudinal sections of ovaries at different days after anthesis (DAA). No bright fluorescence was observed on the ovules at 2 DAA (**a**). By 5 DAA (**b**), fluorescence became visible along the vascular bundles of the ovules, but it disappeared by 8 DAA (**c**); AP = axile placentation, BF = bright fluorescence, O = ovary, Ov = ovule, VB = vascular bundle; Scale bar = 100 µm.

**Figure 4 biology-14-01134-f004:**
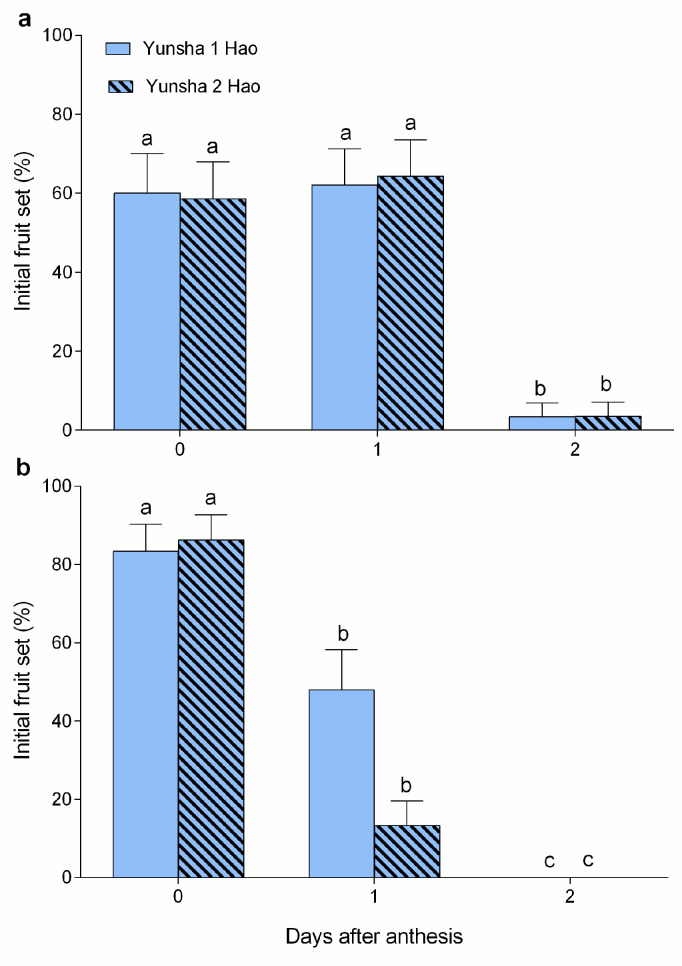
Initial fruit set (%) after pollination at 0–2 days after anthesis (DAA) in either cultivar in 2022 (**a**) and 2023 (**b**). Error bars indicate SE. Different lowercase letters above the error bars indicate significant differences (*p* < 0.05) in IFS among the different pollination dates (presented as DAA) for each cultivar.

**Table 1 biology-14-01134-t001:** The percentages of flowers with pollen adhesion (PA), pollen germination (PG), pollen tube growth in the lower part of the style (LP), and pollen tubes entering ovules (EO) after hand pollination at consecutive numbers of days after anthesis (DAA).

Year ^1^	Cultivar	DAA (days)	PA	PG	LP	EO
2022	Yunsha 1 Hao	0	100.0 ± 0 a ^2^	100.0 ± 0 a	100.0 ± 0 a	95.2 ± 4.8 a
		1	100.0 ± 0 a	100.0 ± 0 a	100.0 ± 0 a	61.5 ± 9.7 b
		2	89.5 ± 7.2 a	63.2 ± 11.4 b	0.0 ± 0 b	0.0 ± 0 c
	Yunsha 2 Hao	0	100.0 ± 0 a	100.0 ± 0 a	90.3 ± 5.4 a	87.1 ± 6.1 a
		1	82.6 ± 8.1 a	78.3 ± 8.8 b	56.5 ± 10.6 b	50.0 ± 10.0 b
		2	100.0 ± 0 a	88.9 ± 11.1 ab	0.0 ± 0 c	0.0 ± 0 c
2023	Yunsha 1 Hao	0	100.0 ± 0 a	100.0 ± 0 a	100.0 ± 0 a	93.3 ± 4.6 a
		1	100.0 ± 0 a	100.0 ± 0 a	96.7 ± 3.3 a	36.7 ± 8.9 b
		2	90.0 ± 5.6 a	73.3 ± 8.2 b	60.0 ± 9.1 b	0 ± 0 c
	Yunsha 2 Hao	0	100.0 ± 0 a	100.0 ± 0 a	100.0 ± 0 a	96.7 ± 3.3 a
		1	100.0 ± 0 a	100.0 ± 0 a	80.0 ± 7.4 b	70 ± 8.5 b
		2	40.3 ± 9.2 b	40.0 ± 9.1 b	30.0 ± 8.5 c	6.7 ± 4.6 c

^1^ Flowers were sampled at 24 h after pollination in 2022 and at 48 h after pollination in 2023. ^2^ Data are presented as mean ± SE. Data points among different pollination days after anthesis in the same column for the same cultivar and year, which are followed by different lowercase letters, are significantly different (*p* < 0.05).

**Table 2 biology-14-01134-t002:** Percentage of flowers with pollen tubes reaching the ovules at different sampling times after pollination.

Sampling Time after Pollination (h)	2022		2023	
	Yunsha 1 Hao	Yunsha 2 Hao	Yunsha 1 Hao	Yunsha 2 Hao
8	8.3 ± 5.8 b	12.0 ± 6.6 c	3.2 ± 3.2 b	0 ± 0 c
24	88.9 ± 6.2 a	75.9 ± 8.1 b	100 ± 0 a	64.5 ± 8.7 b
48	90.0 ± 5.6 a	95.8 ± 4.2 a	100 ± 0 a	87.5 ± 5.9 a

Data are presented as mean ± SE. Data points among different sampling times after anthesis in the same column for the same cultivar and year, which are followed by different lowercase letters, are significantly different (*p* < 0.05).

**Table 3 biology-14-01134-t003:** Percentage of flowers of different ages (expressed as days after anthesis) containing ovules showing fluorescence emission.

DAA ^1^ (days)	2022		2023	
	Yunsha 1 Hao	Yunsha 2 Hao	Yunsha 1 Hao	Yunsha 2 Hao
0	0 ± 0 c ^2^	0 ± 0 c	0 ± 0 c	0 ± 0 c
1	0 ± 0 c	0 ± 0 c	0 ± 0 c	0 ± 0 c
2	0 ± 0 c	0 ± 0 c	0 ± 0 c	0 ± 0 c
3	75.0 ± 13.1 a	36.4 ± 15.2 ab	13.3 ± 6.3 c	0 ± 0 c
4	37.5 ± 18.3 ab	50.0 ± 15.1 a	67.8 ± 6.1 b	92.9 ± 5.0 a
5	57.1 ± 13.7 a	20 ± 13.3 abc	96.4 ± 3.6 a	0 ± 0 c
6	56.3 ± 12.8 a	11.8 ± 8.1 bc	93.5 ± 4.5 a	23.3 ± 7.9 b
7	—	0 ± 0 c	88.0 ± 6.6 a	17.2 ± 7.1 b
8	21.4 ± 11.4 bc	—	58.8 ± 8.6 b	—
9	—	—	—	6.9 ± 4.8 bc

^1^ DAA, days after anthesis. ^2^ Data are presented as mean ± SE. Data points representing different ages (presented as days after anthesis) in the same column for the same cultivar and year, which are followed by different lowercase letters, are significantly different (*p* < 0.05).

**Table 4 biology-14-01134-t004:** Durations (as DAA and number of days) of stigma receptivity (SR), style suitability (SS), pollen tube growth rate (PTG), ovule longevity (OL), and estimated effective pollination period (EPP) of both cultivars in two years.

Year	2022				2023			
Cultivar	Yunsha 1 Hao		Yunsha 2 Hao		Yunsha 1 Hao		Yunsha 2 Hao	
	DAA	Days	DAA	Days	DAA	Days	DAA	Days
SR	0, 1, 2	>3	0, 1, 2	>3	0, 1, 2	>3	0, 1, 2	>3
SS	0, 1	2	0, 1	2	0, 1	2	0, 1	2
PTG		1		1		1		1
OL	0, 1, 2	>3	0, 1, 2	>3	0, 1, 2	>3	0, 1, 2	>4
OL minus PTG		>2		>2		>2		>3
EPP	0, 1	2	0, 1	2	0, 1	2	0, 1	2

## Data Availability

Data are available within the article and Appendix A.

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
