# Peer review of "An Assessment of the Effective Pollination Period and Its Main Limiting Factor in Wurfbainia villosa var. villosa (Lour.) Škorničk. & A. D. Poulsen (Zingiberaceae)"

_biology, 2025, doi:10.3390/biology14091134_

Round 1

Reviewer 1 Report

Comments and Suggestions for Authors

Dear Author: Please delete the lines 323-326 in page 10.

Congratulations for your work.

Author Response

Thank you very much for taking the time to review this manuscript. Please find the detailed responses below and the corresponding revisions in the re-submitted files.

Comments : Dear Author: Please delete the lines 323-326 in page 10.

Response : Thank you for pointing this out. we have delete the lines 323-326 in page 10.

Reviewer 2 Report

Comments and Suggestions for Authors

Dear authors, 

 The MS is a well designed MS with a broad perspective for an important pharmaceutic plant. The EPP is the most important problem in fertilization problems and it is a challenging survey. So, Congratulations. There are some suggestions below for your MS. Please consider them for a better understanding and impressive voice. 

  1. The flower biology of the studied species should be described in ıntroduction section. So that the materials section would be more understandable in terms of why there is not any emasculation procedure. Nobody have to know this pharmaceutic plant.
  2. Please note if there is an emasculation period.
  3. In line 226 the sampling period was written like 48-h but in whole MS it was 48 h (not with a dash). Please make it uniform for whole MS
  4. Line 323-326 is not necessary. I think it is forgotten.
  5. The first paragraph of the discussion section was selected wrong. There have to be a more impressive paragraph. MS is starting with an obstacle and the MS lose its confidence. Please take obstacles of this MS to lower parts of discussion. Because of the style limitation the method is correct. You can say that if there was not a style limitation, the method should be like bla bla..
  6. The authors suggest a new EPP calculation method. This method could be suitable for a plant species which has a style limitation. But for other species the EPP calculation method of Williams is true for most of plant species. Please note that “if there is a limitation like style or something like this”

7. Please check the first reference is it Selak or Vulletin Selak. I used this reference as Vuletin Selak in one of my MS.

Author Response

Thank you very much for taking the time to review this manuscript. Please find the detailed responses below and the corresponding revisions in the re-submitted files.

Comments 1: The flower biology of the studied species should be described in ıntroduction section. So that the materials section would be more understandable in terms of why there is not any emasculation procedure. Nobody have to know this pharmaceutic plant.

Response1: We thank the reviewer for this valuable suggestion. As recommended, we have added a description of the floral biology of W. villosa var. villosa in the introduction section (lines 85-91, p. 2; highlighted in yellow). This addition provides important context for understanding our experimental approach, particularly regarding the absence of emasculation procedures. These biological characteristics are indeed relevant for interpreting our methodology and results.

Comments 2: Please note if there is an emasculation period.

Response2: We appreciate your comment. Based on the floral biology of W. villosa var. villosa added in the introduction (Lines 85–91, p. 2, highlighted in yellow), we know that the anther provides upward growth support for the style, and the stigma is positioned higher than the pollen, which makes self-pollination difficult. Consequently, emasculation is unnecessary under natural conditions.

Comments 3: In line 226 the sampling period was written like 48-h but in whole MS it was 48 h (not with a dash). Please make it uniform for whole MS.

Response 3: Thank you for pointing this out. We have revised "48-h" to "48 h" (line 244, p. 6; highlighted in yellow).

Comments 4: Line 323-326 is not necessary. I think it is forgotten.

Response4: Thank you for pointing this out. Line 323-326 has been deleted.

Comments 5: The first paragraph of the discussion section was selected wrong. There have to be a more impressive paragraph. MS is starting with an obstacle and the MS lose its confidence. Please take obstacles of this MS to lower parts of discussion. Because of the style limitation the method is correct. You can say that if there was not a style limitation, the method should be like bla bla..

Response5: We appreciate the reviewer’s constructive feedback. As suggested, we have restructured the discussion section to avoid beginning with methodological limitations, ensuring a more confident and impactful opening. The first paragraph now emphasizes the key findings and their significance (lines 343–356, p. 10; highlighted in yellow), especially its implications for current production practices. The discussion of stigma receptivity in paragraphs 2–3 (lines 357–362, 367, and 377–378, p. 10; highlighted in yellow) and style suitability in paragraph 4 (lines 379–382, p. 10; highlighted in yellow) have been revised slightly in the discussion.

Comments 6: The authors suggest a new EPP calculation method. This method could be suitable for a plant species which has a style limitation. But for other species the EPP calculation method of Williams is true for most of plant species. Please note that “if there is a limitation like style or something like this”

Response6: We appreciate the reviewer's insightful comment. We agree that our proposed EPP calculation method is particularly suitable for plant species with style limitations, while Williams' method remains valid for most other species. As suggested, we have modified the text to clarify this distinction (lines 335-336, p. 11, highlighted in yellow).

Comments 7: Please check the first reference is it Selak or Vulletin Selak. I used this reference as Vuletin Selak in one of my MS.

Response7: Thank you for pointing this out. We have revised "Selak" to "Vuletin Selak" in the first reference, and more similar revisions were made in the first and sixth references (lines 485, and 494-495, p. 13; highlighted in yellow).

Reviewer 3 Report

Comments and Suggestions for Authors

The article An Assessment of the Effective Pollination Period and its Main Limiting Factor in Wurfbainia villosa var. villosa (Lour.) Skornick. & A. D. Poulsen (Zingiberaceae) by authors Qianxia Li, Yanqian Wang, Ge Li, Shuang Li, Hongyou Zhao, Chunyong Yang, Zhibing Guan, Yating Zhu, Lin Xiao, Yanfang Wang, Lixia Zhang examines the aspects causing low fruit set, associating it with various properties of the generative organs.
The manuscript is formatted according to the rules and contains the necessary sections. The structure of the material and its discussion requires modification.
Main problems
The authors did not formulate the goals, objectives and hypotheses in the last paragraph of the introduction, which is why it is not possible to assess whether the goal has been achieved, this should be corrected.
The authors provide factual data on damage to a particular process, but do not conduct experimental work to eliminate it to prove their claims, and there are no explanations or discussions about the causes of these effects and ways to overcome them. This should be analyzed and corrected.
The reasons may be various morphological changes, differences in pH, chemical composition, the absence or disruption of the formation of polysaccharide lubricant in the pistil, and many other reasons. This should be logically analyzed and offered in the form of a diagram or drawing. Unclear (unexplored options) should be reflected as requiring further study.
The conclusion should reflect the usefulness, novelty of the data obtained by the authors and the prospects and route map for finding solutions.
Minor comments
Any images of plants should be provided with a scale bar - Figure 1.
The journal Biology is an interdisciplinary journal, for this reason it is good practice to have captions on the images, mark the pollen tubes, pistil, membranes, ovary with arrows. If there are no more accurately taken photographs, place a diagram next to it so that it is clear what has changed, where the shells are, where the pollen is, etc.
The expression "bright blue-green fluorescence" should be formulated more scientifically, since it has a range used for analysis.
Conclusions should be more specific.
You recommend a specific study protocol for what.
How your study can improve "crop management". Please avoid general phrases and bring the goals/tasks/hypotheses in line with the conclusions. Particular attention should be paid to conveying novelty in the text, discussion, and conclusions.

Author Response

Thank you very much for taking the time to review this manuscript. Please find the detailed responses below and the corresponding revisions in the re-submitted files.

Main problems
Comments 1: The authors did not formulate the goals, objectives and hypotheses in the last paragraph of the introduction, which is why it is not possible to assess whether the goal has been achieved, this should be corrected.

Response1: We appreciate the reviewer's constructive feedback. As suggested, we have:

  1. Restructured the last paragraph of the introduction section (lines 85–98, and 105–109, p. 2–3; highlighted in yellow) to clearly statethe specific research problem, the primary research goal and its significance, especially its implications for current production practices.
  2. Revised the opening paragraph of the discussion section (lines 343–356, p. 10; highlighted in yellow) toemphasize key findings and their significance, explicitly demonstrate how the findings can be applied in the current production practices.

Comments 2: The authors provide factual data on damage to a particular process, but do not conduct experimental work to eliminate it to prove their claims, and there are no explanations or discussions about the causes of these effects and ways to overcome them. This should be analyzed and corrected.
The reasons may be various morphological changes, differences in pH, chemical composition, the absence or disruption of the formation of polysaccharide lubricant in the pistil, and many other reasons. This should be logically analyzed and offered in the form of a diagram or drawing. Unclear (unexplored options) should be reflected as requiring further study.

Response2: We appreciate the reviewer's constructive feedback. As suggested, we have added explanations in the final paragraph of the discussion section (lines 439–442, page 11; highlighted in yellow) to address the likely causes of these differences and potential solutions.

Comments 3: The conclusion should reflect the usefulness, novelty of the data obtained by the authors and the prospects and route map for finding solutions.

Response3: We appreciate the reviewer's constructive feedback. As suggested, we have added key findings and their significance, explicitly demonstrating how the findings can be applied to current production practices (lines 448–449, and 452-454,page 11; highlighted in yellow).

Minor comments
Comments 4: Any images of plants should be provided with a scale bar - Figure 1.
The journal Biology is an interdisciplinary journal, for this reason it is good practice to have captions on the images, mark the pollen tubes, pistil, membranes, ovary with arrows. If there are no more accurately taken photographs, place a diagram next to it so that it is clear what has changed, where the shells are, where the pollen is, etc.

Response4: We appreciate the reviewer's constructive feedback. As suggested, we have added a more accurate photograph to figure 1, marked the anther, lip, ovary, ovule, pollen, stigma, and style with arrows, and provided a scale bar to all panels (Figure 1, page 11; highlighted in yellow).

Comments 5: The expression "bright blue-green fluorescence" should be formulated more scientifically, since it has a range used for analysis.

Response5: Thank you for pointing this out. We have revised "bright blue-green fluorescence" to "bright fluorescence" in whole article(line 286-287, 291,293,p. 7; line 312,p. 8; line 409,420,p. 11;highlighted in yellow).

Comments 6: Conclusions should be more specific.
You recommend a specific study protocol for what.
How your study can improve "crop management". Please avoid general phrases and bring the goals/tasks/hypotheses in line with the conclusions. Particular attention should be paid to conveying novelty in the text, discussion, and conclusions.

Response6:  We appreciate the reviewer's constructive feedback. As suggested, we have added key findings and their significance, explicitly demonstrating how the findings can be applied to current production practices (lines 448–449, and 452-454,page 11; highlighted in yellow).

Round 2

Reviewer 3 Report

Comments and Suggestions for Authors

The article An Assessment of the Effective Pollination Period and its Main Limiting Factor in Wurfbainia villosa var. villosa (Lour.) Skornick. & A. D. Poulsen (Zingiberaceae) by authors Qianxia Li, Yanqian Wang, Ge Li, Shuang Li, Hongyou Zhao, Chunyong Yang, Zhibing Guan, Yating Zhu, Lin Xiao, Yanfang Wang, Lixia Zhang was edited and corrected.
After the additions and corrections were made, it remains to clarify the wavelength of the studied object and, if possible, make the histograms colored. I also recommend putting designations or arrows on the photographs so that the reader understands where and what tissue is located. Also, in Fig. 3, it is necessary to indicate the type of section (longitudinal, transverse or other) on the pressed preparations, details are also indicated. These minor comments do not prevent the publication of this manuscript and after correction it can be published.

Author Response

Comments : After the additions and corrections were made, it remains to clarify the wavelength of the studied object and, if possible, make the histograms colored. I also recommend putting designations or arrows on the photographs so that the reader understands where and what tissue is located. Also, in Fig. 3, it is necessary to indicate the type of section (longitudinal, transverse or other) on the pressed preparations, details are also indicated. These minor comments do not prevent the publication of this manuscript and after correction it can be published.

Response: We appreciate the reviewer's constructive comments. We have carefully revised the manuscript according to the suggestions. The specific modifications are as follows:

Regarding the clarification of wavelength: The wavelengths used for the studied object have been explicitly stated in the Materials and Methods section (lines 143-144, p. 3; highlighted in green).

Regarding the colored histograms: the only histogram ( in Fig. 4) has been revised to use a color scheme that is clear and distinguishable (lines 337, p. 9).

Regarding the designations on photographs: As suggested, arrows and labels ( "germinated pollen", "ovule","ovary","style","nucellus", "integument", and so on) have been added to the relevant regions in both Fig. 2 (lines 213-219, p. 5-6;highlighted in green)and Fig. 3 ( lines 310-315, p. 8; highlighted in green)to guide the reader .

Regarding the section type and details: The type of section (longitudinal section) has now been clearly indicated in the caption of Fig. 3. Furthermore, additional details regarding the sample preparation and observed structures have been included in the captions as recommended.

Additionally, the labels in Figure 1, which were not alphabetized in the previous version, have now been arranged in alphabetical order (line 157, p. 4; highlights in green). Corresponding adjustments have also been made in the Abbreviations section (line 485, p. 12-13; highlights in green).

We believe that these revisions have significantly improved the clarity and quality of our manuscript. Thank you again for your valuable advice.